# Cost and statistical efficiency of posture assessment by inclinometry and observation, exemplified by paper mill work

**Svend Erik Mathiassen**[1]*, **Amanda Waleh Åström**[1], **Annika Strömberg**[2], **Marina Heiden**[1]

**1** Centre for Musculoskeletal Research, Department of Occupational Health Science and Psychology, Faculty of Health and Occupational Studies, University of Gävle, Gävle, Sweden, **2** Department of Business and Economic Studies, Faculty of Education and Business Studies, University of Gävle, Gävle, Sweden

* svenderik.mathiassen@hig.se

**Data Availability Statement:** All relevant data are within the paper and its Supporting Information files.

## Abstract

Postures at work are paramount in ergonomics. They can be determined using observation and inclinometry in a variety of measurement scenarios that may differ both in costs associated with collecting and processing data, and in efficiency, i.e. the precision of the eventual outcome. The trade-off between cost and efficiency has rarely been addressed in research despite the obvious interest of obtaining precise data at low costs. Median trunk and upper arm inclination were determined for full shifts in 28 paper mill workers using both observation and inclinometry. Costs were estimated using comprehensive cost equations; and efficiency, i.e. the inverted standard deviation of the group mean, was assessed on basis of exposure variance components. Cost and efficiency were estimated in simulations of six sampling scenarios: two for inclinometry (sampling from one or three shifts) and four for observation (one or three observers rating one or three shifts). Each of the six scenarios was evaluated for 1 through 50 workers. Cost-efficiency relationships between the scenarios were intricate. As an example, inclinometry was always more cost-efficient than observation for trunk inclination, except for observation strategies involving only few workers; while for arm inclination, observation by three observers of one shift per worker outperformed inclinometry on three shifts up to a budget of €20000, after which inclinometry prevailed. At a budget of €10000, the best sampling scenario for arm inclination was 2.5 times more efficient than the worst. Arm inclination could be determined with better cost-efficiency than trunk inclination. Our study illustrates that the cost-efficiency of different posture measurement strategies can be assessed and compared using easily accessible diagrams. While the numeric examples in our study are specific to the investigated occupation, exposure variables, and sampling logistics, we believe that inclinometry will, in general, outperform observation. In any specific case, we recommend a thorough analysis, using the comparison procedure proposed in the present study, of feasible strategies for obtaining data, in order to arrive at an informed decision support.

**Funding:** This work was supported by the Swedish Research Council for Health, Working Life and Welfare (Forte Dnr. 2009-1761, recipient SEM; and Forte Dnr. 2010-0748, recipient MH), and the University of Gävle (recipients SEM, AÅ, AS and MH). The funders had no role in study design, data collection and analysis, decision to publish, or preparation of the manuscript.

**Competing interests:** The authors have declared that no competing conflicts interests exist.

## Introduction

Musculoskeletal disorders (MSDs) among workers are a major obstacle for arriving at a sustainable working life [1]. Working postures of, e.g. the trunk and upper arm are considered important to the risk of getting disorders [2], and a large number of studies have assessed and interpreted postures in different occupations using a variety of methods, including self-reports, observations and direct measurements [3, 4]. Irrespective of the measurement method, data need to be accurate, i.e. both true and precise [5]; both in research, since accuracy is a decisive determinant of the ability of an epidemiologic study or an intervention to give useful evidence [6], and in practice, since documentation, surveillance and interpretation of postures need to be sensitive to modifications that may occur as a result of changes in production [7]. While lack of trueness can, in many cases, be adjusted for [8–10], precision refers to unexplained variance in results that can be handled only through the sampling strategy, i.e. the method used for collecting and processing data. A number of studies have addressed factors influencing precision in sampling, such as the performance of different exposure assessment methods [11–14]; sizes of exposure variance components between and within subjects [15–18]; the amount of collected data [19]; effects of distributing measurements differently between and within subjects, and within days [20, 21]; the ability of standard sampling strategies to detect the effects they were intended to detect [6, 7]; and the efficiency of modeling exposure on basis of, e.g. administrative data [22–25].

Obviously, given a certain method and sampling strategy, more measurements will give a better efficiency, i.e. a more precise estimate of the mean exposure of the investigated population. However, more measurements come at an increased cost, and in most cases the researcher or the practitioner has access only to a limited budget. Thus, both researchers and practitioners face the challenge of identifying a sampling strategy that will either lead to a certain efficiency at the lowest possible cost, or–in reverse–arrive at the best possible efficiency at a certain budget. In extension, both researchers and practitioners will need to address whether the study design allowed by the budget constraints is, indeed, sufficiently informative to be meaningful. The need to consider both costs and efficiency is also an important element of good research practice, expressed by the European Federation of Academies of Sciences and Humanities in their European Code of Conduct for Research Integrity as: 'Researchers *(will need to)* make proper and conscientious use of research funds' [26]. Surprisingly few studies have, however, addressed the trade-off between costs and efficiency in different measurement strategies for different exposure variables. In a 2010 review, Rezagholi and Mathiassen [27] identified only nine studies with explicit analyses of both cost and efficiency. In general, these studies suffered from low quality in the methods used for cost assessment and none of them addressed biomechanical exposures, let alone postures. A few studies of relevance to biomechanical exposures had appeared before 2010, including, e.g. an estimate of the proportion of lost measurements when using inclinometry and observation [28], but the studies did not include an explicit cost model. Some studies of biomechanical exposure assessment have appeared since 2010, showing clearly that a data collection strategy that would appear preferable if only efficiency were considered may not be preferable anymore if costs are also included [29, 30]. Other studies have presented comprehensive cost models, addressing costs during data collection [31] and data processing [32]. Cost and efficiency may differ between posture measurement methods, and observation and direct measurements have been examined, both separately [33], and in a comparative study [34], concluding that observation was more cost-efficient than inclinometry in some sampling scenarios, while the opposite applied to other scenarios. The literature on cost-efficient assessment of biomechanical exposures is, however, still very limited, despite the obvious interest in collecting efficient data at a low price. The

available literature suggests that costs may differ widely depending on the outcome variable (e.g. arm or trunk posture), the measurement method (e.g. inclinometry or observation), and the sampling strategy (e.g. the number of assessed subjects and shifts); and also that the same three factors are decisive for the statistical efficiency of the eventual result. Furthermore, costs of collecting data are steadily changing, e.g. with the technologic development of accelerometers [35], and the cost-efficiency of different approaches for sampling data may need reappraisal at short intervals.

The aim of the present study was to determine and compare the trade-off between cost and statistical efficiency, i.e. the cost-efficiency, for assessments of trunk and upper arm posture using inclinometry and observation. We estimated cost-efficiency for both inclinometry and observation under sampling strategies differing by the number of workers and work-shifts per worker, and–for observation–the number of observers, and exemplified our estimation procedures by data collected in paper mill work.

## Methods

### Study population

Data were collected between March and June 2011 during three full shifts in each of 28 randomly selected workers out of a total of 55 employees at a Swedish paper mill [17, 23]. Work tasks consisted of operating paper machines, testing paper quality, transporting and packaging paper rolls, and monitoring and/or controlling the production. Workers were eligible into the original study if they had a full-time job with no modified duties. Since we wished to focus the present study on inclinometry and observation of manual work, we excluded shifts in which the worker reported to spend more than 50% of his/her time in computer work. This procedure led to the total number of eligible shifts being reduced to 66, and 4 workers being entirely disqualified.

All participants at the paper mill were informed about the study both verbally and in writing and signed an informed consent to participate. The study was conducted in accordance with the declaration of Helsinki and approved by the Regional Ethical Review Board in Uppsala (2011/026). One of the authors (MH) had access to individualized information on the study participants, but did not share this, neither with the other authors nor with persons not involved in the present study.

### Posture assessment

Trunk and upper arm postures were measured for each worker and shift, using both inclinometry and observation from videos. A detailed description of the data collection can be found in previous papers [17, 23]. In brief, three inclinometers were placed on the worker in the morning of a work shift; one on the trunk between the shoulder blades, and two over the medial deltoids of the right and left upper arms. The worker then wore the inclinometers for the entire shift, after which they were removed. This procedure was repeated for three shifts, intended to reflect different shift types, e.g. in terms of duration. Zero inclination reference postures were determined for each shift using established techniques [17]. During the three work shifts, a camera operator from the research team also followed the worker, aiming at capturing the trunk and right upper arm on video. Afterwards, three observers estimated trunk and upper arm postures relative to vertical from video still frames at 135s intervals, aided by a customized software [17, 23]. The videos were also used to identify working hours in the inclinometry recordings. Both field data collectors and observers received standardized training before collecting and processing the data [32]. Trunk and right upper arm inclination angles determined by inclinometers and estimated by video observation were further processed to obtain the median trunk and right upper arm inclination during each shift. Due to technical problems, trunk inclinometry

was lost for one shift, and arm inclinometry for two. Thus, the present study was based on data regarding median trunk and right upper arm inclination from, in total, 66, 65 and 64 shifts (observation, trunk inclinometry, and arm inclinometry, respectively) in 24 workers.

## Estimation of costs

Time spent by researchers during the complete planning and implementation of the paper mill study was meticulously documented for each task (*e.g.*, recruiting subjects, setting up and performing data collection, data processing), allowing costs to be assessed according to the model proposed by Trask et al. [31, 32, 34] and modified by Waleh Åström et al. [36]:

$$C_{Tot} = \check{C}_E + \check{C}_R + \check{C}_S + \check{C}_T + \dot{C}_d + \dot{C}_m + \dot{C}_{R-group} + \dot{C}_{R-ind} + \dot{C}_v + \dot{C}_h + \dot{C}_a \qquad (1)$$

This general cost model includes fixed cost components for equipment, recruitment of organization to study, software development, and training ($\check{C}_E, \check{C}_R, \check{C}_S, \check{C}_T$, respectively); as well as variable cost components associated with data acquisition, data processing, recruitment of workers in groups and individually, travelling to the worksite, overnight accommodation, and administration, ($\dot{C}_d, \dot{C}_m, \dot{C}_{r-group}, \dot{C}_{r-ind}, \dot{C}_v, \dot{C}_h, \dot{C}_a$, respectively). Fixed costs are defined as costs that do not depend on the size of the study. Variable costs, on the other hand, do depend on the size of the study, and are calculated by multiplying a unit cost, i.e. the cost associated with retrieving one measurement unit, by the number of units collected in the study. In any specific case, this general cost equation can be operationalized by adding up the fixed cost components, i.e. $\check{C}_E, \check{C}_R, \check{C}_S,$ and $\check{C}_T$, to a summary variable, i.e. $\check{C}_F$, and complementing that by a number of equations expressing the effects of different sources of variable costs. In the present study, we operated with five measurement units, i.e.:

- *groups of workers* (unit cost $\dot{c}_g$) for costs associated with explaining the study to groups of 10 workers at a time (overall cost $\dot{C}_{r-group}$);

- *workers* (unit cost $\dot{c}_w$) for costs associated with recruiting individual workers ($\dot{C}_{r-ind}$);

- *shifts* (unit cost $\dot{c}_d$) for travels to the worksite for both inclinometry and observation ($\dot{C}_v$), for attaching and detaching inclinometers and for processing inclinometer data ($\dot{C}_d$ and $\dot{C}_m$ for inclinometry), and for data collection in observation ($\dot{C}_d$ for observation);

- *observations* (unit cost $\dot{c}_o$) for one post-hoc assessment by one observer of postures in a video recording from an entire shift ($\dot{C}_m$ for observation).

- Costs associated with *administration* ($\dot{C}_a$, unit cost $\dot{c}_a$) were calculated as an overhead on all other costs and set at 24% and 15% for inclinometry and observation, respectively [36].

The present study did not entail any costs for overnight accommodations ($\dot{C}_h$) since the work site was close to the residences of involved researchers [36].

Thus, the total cost of collecting and processing posture data based on inclinometry ($C_{Tot, inc}$), and observation ($C_{Tot,obs}$) was calculated as:

$$C_{Tot,inc} = [\check{C}_F + \dot{c}_g n_g + \dot{c}_w n_w + \dot{c}_d n_w n_d](1 + \dot{c}_a) \qquad (2A)$$

and:

$$C_{Tot,obs} = [\check{C}_F + \dot{c}_g n_g + \dot{c}_w n_w + \dot{c}_d n_w n_d + \dot{c}_o n_w n_d n_o](1 + \dot{c}_a) \qquad (2B)$$

where:

- $\check{C}_F$ is the total fixed cost, i.e. the sum of $\check{C}_E$, $\check{C}_R$, $\check{C}_S$ and $\check{C}_T$

- $\dot{c}_g$, $\dot{c}_w$, $\dot{c}_d$ and $\dot{c}_o$ are the unit costs for *groups of workers*, *workers*, *shifts* and *observations*, respectively

- $\dot{c}_a$ is the administrative overhead, expressed as a proportion

- $n_g$, $n_w$, $n_d$, and $n_o$ are the number of groups, workers, shifts per worker, and observers, respectively. Observers observed each video only once.

Fixed costs and unit costs were retrieved from the material reported by Waleh Åström et al. [36].

## Estimation of statistical performance

Statistical performance of the measurement methods, i.e. their efficiency, was estimated on the basis of variance components derived using models modified from Trask et al. [34, 37].

For inclinometry:

$$y_{ij} = \mu + \beta_i + \varepsilon_{ij} \tag{3A}$$

For observation:

$$y_{ijk} = \mu + \beta_i + \beta_{j(i)} + \beta_k + \beta_{ik} + \varepsilon_{ijk} \tag{3B}$$

where:

- $y_{ij}$ is the posture for worker i in shift j, according to inclinometry

- $y_{ijk}$ is the posture for worker i in shift j, as assessed by observer k

- $\mu$ is the grand mean

- $\beta_i$ is the random effect of worker with i = 1, 2,.., 28

- $\beta_{j(i)}$ is the random effect of work shift nested within worker, with j = 1, 2, 3

- $\beta_k$ is the random effect of observer, with k = 1, 2, 3

- $\beta_{ijk}$ is the interaction between worker and observer

- $\varepsilon_{ij}$ and $\varepsilon_{ijk}$ are residual error effects, representing, for inclinometry the effect of shifts within worker, and, for observation combined residual effects within observers, workers and work-shifts.

The models were resolved in SPSS 22.0 for Windows (IBM Corp., Armonk, NY, USA) using restricted maximum likelihood (REML) algorithms [38] in a Varimax procedure to give the variance components associated with workers ($S^2_{BW}$), work shifts within workers ($S^2_{WW,}$), observers ($S^2_{BO}$), interaction between workers and observers ($S^2_{BWBO}$), and a residual ($S^2_{res}$), reflecting variability in observed postures within observers and work-shifts. The within-observer variability could not be isolated in the present study, since observers did not repeat their observations of the videos [33].

The estimated variance components were then used to assess the precision, in terms of the standard deviation (SD), of a group mean posture for different sampling strategies (cf. section 'Sampling strategies'). Statistical performance (efficiency) of inclinometry, $Q_{inc}$, and observation, $Q_{obs}$, was expressed in terms of 1/SD, where a larger 1/SD reflects a better efficiency, i.e. a more precise estimate of the group mean [33, 34].

Thus, for inclinometry:

$$Q_{inc} = 1/\sqrt{(S_{BW}^2/n_w + S_{WW}^2/n_w n_d)} \tag{4A}$$

and for observation:

$$Q_{obs} = 1/\sqrt{(S_{BW}^2/n_w + S_{WW}^2/n_w n_d + S_{BO}^2/n_o + S_{BWBO}^2/n_w n_o + S_{res}^2/n_w n_d n_o)} \tag{4B}$$

where:

- $S_{BW}^2$, $S_{WW}^2$, $S_{BO}^2$, $S_{BWBO}^2$, $S_{res}^2$ are variance components as explained above

- $n_w$, $n_d$, and $n_o$ are the number of workers, shifts per worker, and observers, respectively.

Eqs 4A and 4B both follow from standard procedures for transforming the variance components in a specific study design (derived using Eqs 3A and 3B) into the expected statistical performance of a corresponding estimate of a group mean exposure [39].

## Sampling strategies

Costs and statistical efficiency were estimated for sampling strategies differing in the number of workers ($n_w$ = 1 to 50), the number of shifts per worker ($n_d$ = 1 or 3) and, for observation, the number of observers ($n_o$ = 1 or 3). For each of the 100 combinations of $n_w$ and $n_d$ in inclinometry, costs and efficiency were estimated using Eqs 2A and 4A, respectively. Similarly, Eqs 2B and 4B were used to estimate costs and efficiency for each of the 200 combinations of $n_w$, $n_d$ and $n_o$ in observation. These, in total, 300 estimates of costs and statistical efficiency were derived for both trunk and upper arm inclination.

## Results

### Variance and cost components

As described above, cost components for Eqs 2A and 2B (Table 1) were retrieved on basis of materials reported in a previous publication [36].

Variance components required for Eqs 4A and 4B were obtained by analyzing inclinometry and observation data from the paper mill (Table 2).

### Cost vs. statistical efficiency

Fig 1A (upper arm inclination) and 1b (trunk inclination) illustrate the trade-off between cost and statistical efficiency for the investigated sampling strategies.

**Table 1. Cost components (€) in inclinometry and observation.**

| Cost component | Inclinometry | Observation |
|---|---|---|
| $\breve{C}_F$ | 3465.6 | 3644.9 |
| $\dot{c}_g$ | 86.2 | 86.2 |
| $\dot{c}_w$ | 5.9 | 5.9 |
| $\dot{c}_d$ | 112.1 | 310.3 |
| $\dot{c}_o$ | - | 20.9 |
| $\dot{c}_a$ | 0.24 | 0.15 |

$\breve{C}_F$, fixed costs; $\dot{c}_g$, unit cost, groups of workers; $\dot{c}_w$, unit cost, workers; $\dot{c}_d$, unit cost, shifts; $\dot{c}_o$, unit cost, observation; $\dot{c}_a$, administrative overhead

**Table 2. Group mean values and variance components of the median inclination of the upper right arm and trunk according to inclinometry and observation.**

| | Inclinometry | | Observation | |
|---|---|---|---|---|
| | Upper arm | Trunk | Upper arm | Trunk |
| Group mean,° | 30.9 | 4.3 | 18.5 | 2.4 |
| Variance component,°$^2$ | | | | |
| $S^2_{BW}$ | 28.4 | 10.1 | 8.0 | 8.1 |
| $S^2_{WW}$ | 22.7 | 64.2 | 2.4 | 3.3 |
| $S^2_{BO}$ | - | - | 1.1 | 6.6 |
| $S^2_{BWBO}$ | - | - | 14.7 | 3.5 |
| $S^2_{res}$ | - | - | 19.9 | 4.5 |

$S^2$, variance; BW, between workers; WW, between shifts within worker; BO, between observers; BWBO interaction between workers and observers; res, residual

The two diagrams illustrate the general property of all sampling strategies of a decreasing return to scale, i.e. that an investment in more data (i.e. a more costly study) leads to a still smaller increase in efficiency, the more data is already available. This is a result of the inherent non-linear nature of the equations describing statistical efficiency (Eqs 4A and 4B), while cost increases (almost) linearly with the number of workers for each individual curve, i.e. at a specified number of shifts/worker, and observers (cf. Eqs 2A and 2B). As an example, in trunk observation by 3 observers, 1 shift/worker (Fig 1B, blue filled triangles), increasing the budget from about €5000 to about €10000 leads to a return in terms of increased efficiency (1/SD) from about 0.30°$^{-1}$ to about 0.55°$^{-1}$, while the same investment of €5000 on top of €10000 already invested is accompanied by an increase in efficiency from about 0.55°$^{-1}$ to about 0.60°$^{-1}$; i.e. only one fifth of the return associated with the former investment.

The figures can be used to appraise sampling efficiency at a specified budget, i.e. a specified value on the cost axis. As an example, illustrated by the vertical dashed line in Fig 1A and enlarged in Fig 2A, a budget allowance of €10000 would return an efficiency of 0.84°$^{-1}$ for the best strategy in arm inclinometry, i.e. the sampling strategy closest 'left' to the dashed line (36 workers, 1 shift/worker; red triangles), but only 0.68°$^{-1}$ for the best possible observation strategy (12 workers, 3 observers, 1 shift/worker; blue filled triangles), and down to 0.34°$^{-1}$ for the least efficient observation strategy allowed by the €10000 budget frame (4 workers, 1 observer, 3 shifts/worker; blue open squares). Efficiencies, 1/SD, of 0.84, 0.68 and 0.34°$^{-1}$ correspond to SD-values of 1.19°, 1.47° and 2.94°, respectively. If data were, as an example, normally distributed, these SDs would result in 95% confidence limits on an estimated group mean of median arm inclination of ±2.3°, ±2.9°, and ±5.8°, respectively; i.e. an uncertainty of the estimated mean value which is 2.5 times larger in the least favorable measurement strategy than in the most favorable.

The figures can also be read in terms of the cost of different strategies at a specified statistical efficiency, i.e. a certain value on the 1/SD axis. For instance, a study aiming at determining the group mean trunk inclination in an occupational group to within ±5° (in terms of a 95% confidence interval), would–again assuming normally distributed data–require the SD to be no more than 2.55°, i.e. 1/SD needs to be at least 0.39°$^{-1}$. As illustrated by the horizontal dashed line in Fig 1B, enlarged in Fig 2B, this target efficiency can be obtained at the least cost, €6028, by collecting observation data from 4 workers, 1 shifts/worker, and have 3 observers rate the trunk postures (blue filled triangles), i.e. the closest sampling strategy 'above' the line. The target efficiency can also be reached by inclinometry from a sample of 12 workers, 1 shift/worker (red triangles), but then at a cost of €6262; from inclinometry on 5 workers, 3 shifts/worker (red squares) at a cost of €6521, and from having 3 observers rate video stills from 3 workers, 3 shifts/worker (blue filled squares). The latter strategy would, however, cost €8165; i.e. 35%

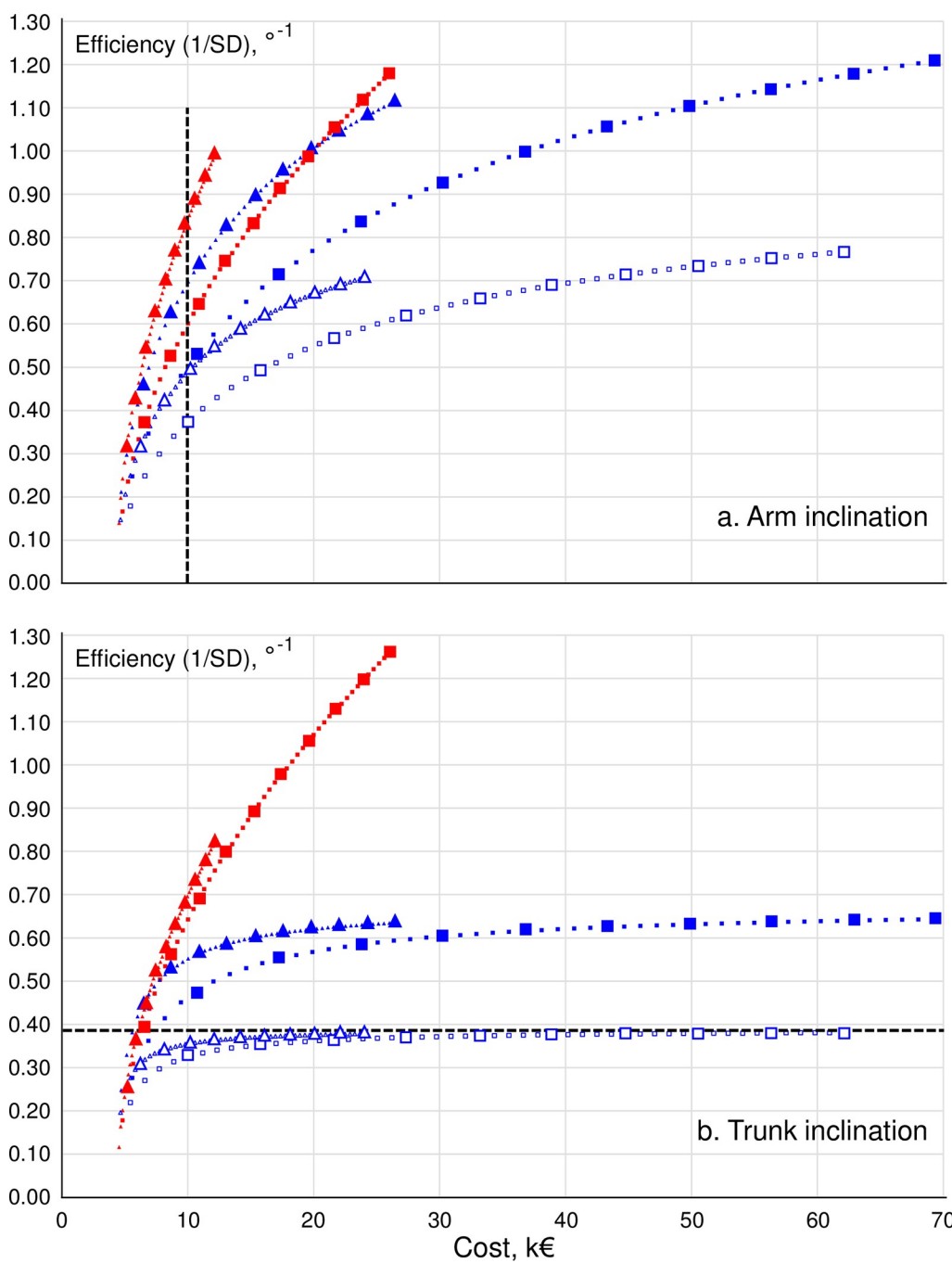

**Fig 1.** Cost and statistical efficiency of upper arm (Fig 1A) and trunk (Fig 1B) inclination assessed by inclinometry (curves in red) and observation (curves in blue). The figure illustrates sampling strategies differing by the number of workers (individual points in each curve, $n_w$ = 1 to 50; large symbols marking $n_w$ = 5, 10, 15, . . ., 50), shifts per worker (triangles, $n_d$ = 1; squares, $n_d$ = 3), and, for observation, observers (open symbols, $n_o$ = 1; closed symbols, $n_o$ = 3). Dashed vertical (Fig 1A) and horizontal (Fig 1B) lines refer to examples explained in the running text.

more than the best strategy offering the same efficiency. Notably, the target efficiency of $0.39^{\circ-1}$ cannot be reached at all in a sample of 50 workers using observation strategies engaging only 1 observer (blue open squares and triangles in Fig 1B; outside the excerpt in Fig 2B).

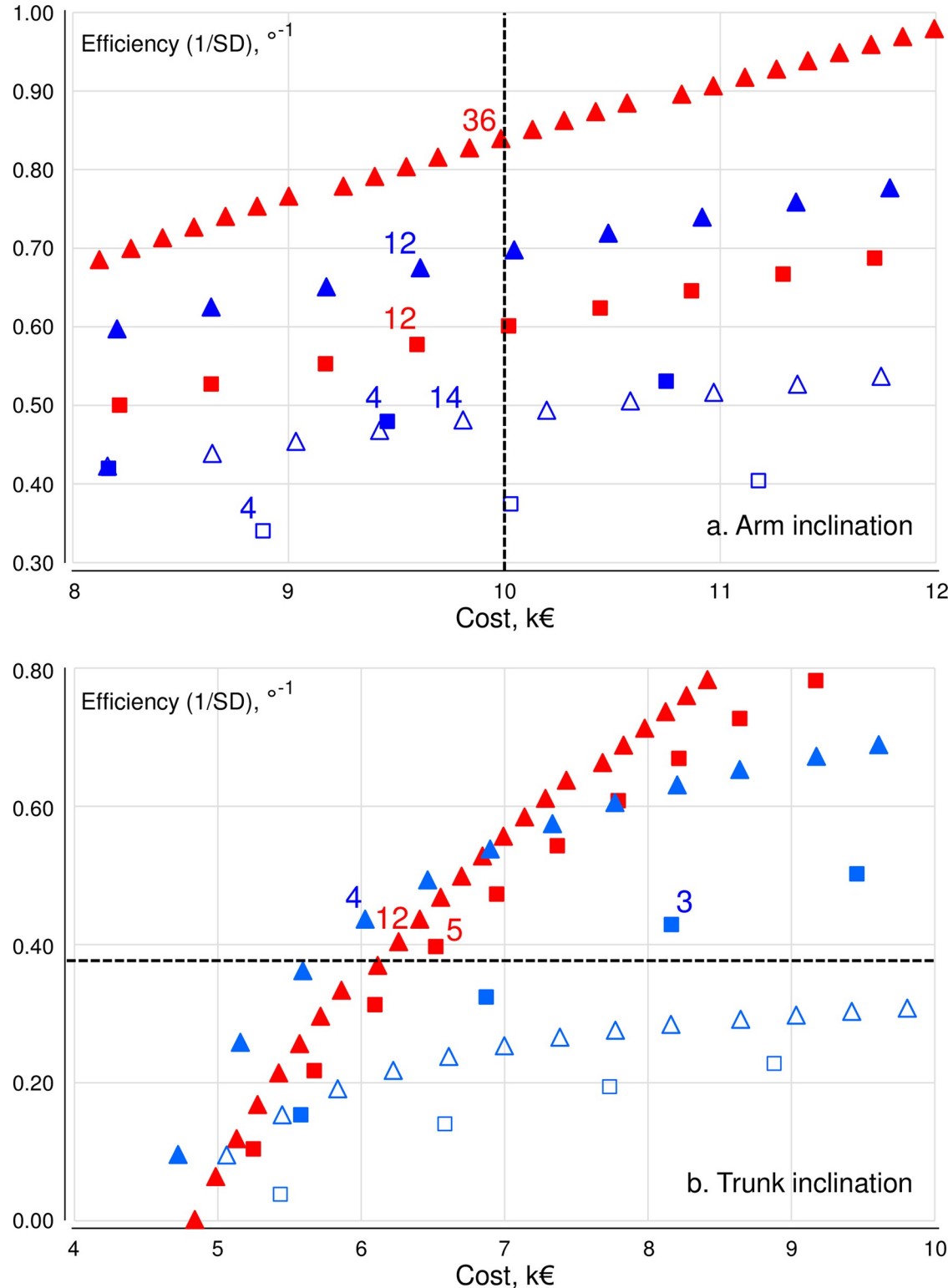

**Fig 2.** Enlarged sections of Fig 1, corresponding to the examples in the running text of sampling efficiency for different strategies at a specified budget (Fig 2A), and costs for different strategies at a specified efficiency (Fig 2B). Numbers inside the figure show the size of studies complying with the criterion in each of the two cases, i.e. 'a cost no larger than €10000' (Fig 2A) and 'an efficiency corresponding to a 1/SD of at least 0.39°⁻¹' (Fig 2B). Red and blue symbols: inclinometry and observation; triangles and squares: one shift and three shifts per worker; open and closed symbols: one and three observers.

Finally, the figures can be read as a basis for comparing the cost-efficiency of any two sampling strategies $p$ and $q$ in terms of their relative efficiency divided by their relative cost; i.e. $Q_p C_q / Q_q C_p$, where $Q_p$ and $Q_q$ are the efficiencies of the two strategies, and $C_p$ and $C_q$ their costs (e.g. [29]). As an example, measuring arm inclination by inclinometry on 25 workers, 1 shift each (Fig 1A, red triangles) will cost €8269 ($C_p$) and leads to an efficiency of 0.70°$^{-1}$ ($Q_p$). Having 3 observers rate arm inclination on the same 25 workers, 1 shift each (Fig 1A, blue filled triangles), results in a better efficiency, i.e. 0.90°$^{-1}$ ($Q_q$), but it is also considerably more expensive: €15369 ($C_q$). The relative cost-efficiency of these two strategies is 1.45; thus, the strategy based on inclinometry is 45% better than that based on observation.

## Discussion

We estimated the association between cost and efficiency for different measurement strategies in observation or inclinometry, using our procedure based on variability between and within workers in median trunk and upper arm inclination at work, and a comprehensive record of costs associated with collecting and processing data. In the present case of trunk inclination in paper mill work, inclinometry (Fig 1B, red symbols) was more cost-efficient than observation (blue symbols), except for some strategies involving 3 observers rating trunk inclination for 1 shift on only few workers (Fig 1B, blue filled triangles; enlarged in Fig 2B). For arm inclination, on the other hand (Fig 1A), inclinometry on 1 shift/worker was always the more cost-efficient (red triangles), while observation by 3 observers of 1 shift/worker (blue filled triangles) was more cost-efficient than the second best inclinometry strategy (red squares) up to a budget of about €20000; at larger budgets inclinometry was better. Arm inclination could be determined with considerably better cost-efficiency than trunk inclination in both inclinometry and observation if the superior measurement strategy was used in all cases.

### Cost and efficiency

Our analyses of costs were based on the cost model proposed and used by Trask et al. [31, 32, 34], with some modifications suggested by Waleh Åström et al. [36]. This comprehensive cost model was presented in response to cost assessments in exposure studies before 2012 being restricted and insufficient in not including all relevant costs (e.g. [40–43]). As a measure of efficiency, we used the reciprocal standard deviation of a group mean exposure value, estimated according to standard equations based on the variance components of the data set (e.g. [6, 21, 33, 34, 37, 44, 45]). We used specific equations for inclinometry and observations, with the difference describing the effect of between-observer variability, i.e. variability in the performance of the measurement instrument (i.e. the observer). Inclinometers may also, in theory, exhibit a variance introduced by the measurement instrument, i.e. a 'between-inclinometer variance'. However, this variance is negligible [46], and we decided to not include it, in keeping with a far majority of inclinometer studies in the past [11, 44, 47–49]. Notably, since this methodological variance component is specific to the observation instrument while other variance components are common to observations and inclinometry, observations can be expected to result in less statistical efficiency when data are collected from a specific number of subjects and shifts.

### Previous research on cost-efficiency

Only one previous study has, to our knowledge, compared the cost-efficiency of observations and inclinometry when assessing working postures of the arm and trunk [34]. That study, using a comprehensive cost model similar to the one used by us, reported that for some posture variables, inclinometry was more cost-efficient than observation, while for other variables it was opposite. The present study found that the best strategy in inclinometry was better than

the best observation strategy for both arm and trunk inclination, but also that observation could outperform less cost-efficient inclinometry, in particular for arm inclination. A number of differences between the studies may explain this discrepancy and may serve to illustrate that the relationship between cost and efficiency will be specific to a particular data collection since the cost of measuring and processing data and the variability of the resulting data set is specific to the investigated population and data collection strategy. As an example, the Trask et al. study [34] entailed larger costs than the present study in terms of overnight housing costs for the researchers, while, on the other hand, inclinometry as practiced by us required inclinometers to be mounted on subjects prior to each day of measurements rather than only once, as in the Trask study [34]. The latter study also presented still frames to observers for every 55s of work, but only for a half shift–in total 252 frames–while we had observers rate still frames for the entire shift, but only every 135s, in total about 215 frames. Thus, while the costs of processing observation data for one observer is slightly less in the present study ($\dot{c}_o$ in Table 1), costs for video recording workers on site is considerably larger (the major part of $\dot{c}_d$ in Table 1). Exposure variance components for median trunk and upper arm angles also differ between the Trask et al. study [34] and the present results. This was expected, since the Trask et al. study was performed on flight baggage handlers and the present study on paper mill workers [17]. The size of variance components and even the relative contribution of different sources of variance to the total depends on the addressed exposure variable [16, 29, 34, 45, 47], and also on the properties of jobs in the investigated occupation, both in terms of the tasks performed in the job and the way tasks are allocated to workers [50].

## Cost-efficient inclinometry and observation

For both inclinometry and observation, cost-efficiency was better in strategies involving 1 shift/worker (red and blue triangles in Fig 1) than in strategies where 3 shifts were assessed per worker (red and blue squares in Fig 1). For instance, having 3 observers observe trunk inclination for 1 shift in each of 30 workers (Fig 1B, blue filled triangles) will cost €17546, while having the same 3 observers observing 3 shifts will cost €43260, i.e. 247% of the cost for 1 shift. However, efficiency differed by only 1% between the two strategies. The cost-efficiency relationship between inclinometry and observation can be understood when inspecting cost and variance components together (Tables 1 and 2). The cost of one additional shift–$\dot{c}_d$ in Table 1 –is larger in observation than in inclinometry, mainly because the cost of video filming is large, while the variance between shifts–$S^2_{WW}$ in Table 2 –is quite small in observation. Thus, adding one shift in observation will entail large costs at the benefit of only a small increase in efficiency [33], compared to inclinometry. For simple cost models, the best possible, i.e. optimal, allocation of measurements to either workers or shifts within workers can be determined analytically [30, 43, 51], but this is not possible with the comprehensive cost model used in our procedure [36]. The complexity of the cost model precludes an analytical optimization of the measurement strategy, and only allows for a comparison of selected alternative strategies, as exemplified in the present study.

The cost-efficiency relationship between inclinometry and observation depended on the available budget. Thus, at a total cost of, for instance, €8000, the best inclinometry strategy for arm inclination, i.e. sampling 1 shift per worker (Fig 1A, red triangles), resulted in an efficiency which was 113% better, i.e. more than double, than that of the worst observation strategy, i.e. having 1 observer observe 3 shifts (Fig 1A, blue open squares). At a budget of €12000, inclinometry instead led to a 145% better efficiency. This illustrates that the relative risk of wasting money by choosing a less favorable method and an inferior data collection strategy can be considerable, and that it may increase if the budget is large. At the same time, arm

inclination data also illustrate that the 'larger budget, larger difference' notion is not always valid, and that inclinometry may not always be superior to observation. Thus, for budgets less than €20000, inclinometry of 3 shifts (Fig 1A, red squares) was less cost-efficient than having 3 observers observe 1 shift (Fig 1A, blue closed triangles), while for budgets larger than €20000, inclinometry was the better. For budgets between €17000 and €25000, the two strategies differed in efficiency by less than 5%.

## Cost-efficiency in general

While the explicit results of the present study (Fig 1) are specific to the investigated population and measurement strategies, and to the Swedish cost structure, they illustrate some general results that can be expected to apply even to other populations. As a trivial finding, common to all cost-efficiency studies using 1/SD as the efficiency metric, data collection exhibits a decreasing return to scale, i.e. that the marginal increase in performance corresponding to adding another subject to an already existing sample gets less the larger the sample [33, 34]. This is a result of the efficiency metric 1/SD increasing as a function of the square root of the number of subjects assessed, while costs develop almost linearly. The study also illustrates that strategies appearing to be preferable when only considering statistical efficiency may not be the most cost-efficient [29]. Thus, a search for effective measurement strategies may be misguided if costs are not properly considered.

The present study, and hence the comparison between inclinometry and observations, is obviously based on the current standings of these two methods and how they were practiced in the present case. Other measurement strategies would likely have changed both efficiency, such as if observers were trained even more to reduce between-observer variance [52], and costs, such as if experienced researchers were hired, with larger salaries but less need for training than the assistants used by us. The effects on cost-efficiency of even more extreme scenarios, including e.g. that all instrumentation is already available in a 'stock', were estimated by Trask et al. [34], showing that this can substantially change the relationship between inclinometry and observation.

By definition, the variable costs increase with the number of subjects and measurements per subject while the fixed costs, i.e. $\check{C}_F$ in Eqs 2A and 2B, do not change. Thus, in small data collections, fixed costs, such as whether instrumentation is available or not, are larger than variable costs, while the opposite holds for larger data sets. In the present case, the turning point where the variable costs overtake the fixed costs is reached for about 25 workers in the best inclinometry strategies, but at the most for only about 10 workers in observations. Thus, changing the fixed costs will have a larger relative effect in inclinometry than in observations. While the fixed costs are, to a large extent, related to purchasing equipment, the variable costs depend on the procedure for sampling and processing data. The fixed costs are, thus, sensitive to technologic development, while the variable costs may change preferentially because of different measurement scenarios.

## Inclinometry and observation in the future

In a near future, observation technologies for upper arm and trunk postures may not change to any major extent. Postures cannot readily be observed from cameras that do not capture the trunk or upper arm, such as for instance recordings made by body-worn cameras. Inclinometry, on the other hand, may become considerably cheaper in the future. As an example, Rezagholi et al. [29] used custom built inclinometers [53] since no devices were commercially available then. The present study and the study by Trask et al. [34] used a more modern technology at a price of about 25% of the previous inclinometers, and to-day, postures can be

measured using commercial inclinometers at 10% the price of those used by us (e.g. [54, 55]). This development in costs allows very large data sets to be collected using inclinometry, and initiatives have even been taken to merge several data sets into joint data bases [56]. Access to cheap inclinometers that are easy to handle–as opposed to those originally used by, e.g. Reza-gholi et al. [29]–also allows for the development of cheaper data collection strategies, where the subject herself is responsible for larger parts of the data collection and data are transferred to the researchers only for further processing and analysis, which may even be automated [57]. In some cases, the subject may even collect and share posture data collected using the in-built accelerometers in her smart phone [58].

Lower fixed costs shift the cost-efficiency curves (cf. Fig 1) to the left, while cheaper data collection procedures reduce the variable cost per measurement unit and are reflected in steeper curves. In a near future, developments of inclinometers and sampling strategies in inclinometry will further increase the relative cost-efficiency advantage of inclinometers, com-pared to observations. However, technological developments may, in a more distant future, lead to automated procedures based on Artificial Intelligence (AI), e.g. for determining pos-tures from single camera views in smart phones. This will change the cost-efficiency relation-ship between inclinometry and observation, and now to the advantage of the latter.

In a further development of exposure assessment strategies, exposures to allegedly 'costly' variables, such as inclinometry or electromyography, can likely be estimated from statistical models based on cheaper data. As an example, studies have proposed exposure prediction models based on self-reports [24], work tasks [22, 59, 60], or administrative information from the company [23]. However, modeling has so far not been very successful in predicting expo-sures, even if some promising examples exist [25], and the cost-efficiency of exposure model-ing has not been systematically evaluated.

## Limitations

**Trueness of observations and inclinometry.** The present study has some limitations. First, we did not address trueness [5], neither in inclinometry nor in observation even though the mean exposure values differed considerably between the methods (Table 2). Discrepancies is a common finding in studies reporting both inclinometry and observation data from the same occupation [11, 14, 17, 29, 34, 61, 62]. As demonstrated by Trask et al. [34], the relative cost-efficiency of the two methods depends profoundly on which one is considered to reflect the 'true' exposure. Conventionally, inclinometry has been used as the 'gold standard', and deviations from inclinometry data in observational studies have been interpreted to indicate that observations are less valid [3]. However, this idea has been challenged [9] and we find that it would be premature to select either inclinometry or observation to reflect the 'truth'.

**Lost data.** We also assumed that all measurement efforts eventually contributed to the estimates of mean inclinations. However, both inclinometry and observations are associated with loss of data in field studies. Inclinometry may, for instance, fail due to technical problems with the sensors, or because the work task interferes with the inclinometers, and the loss may amount to about 10% of the intended data [28]. Observations may be compromised for instance by sequences of the job not being fully visible [63], and data during these periods may either deviate systematically from uncompromised observations, be more uncertain or be completely lost [37]. In the present data material, 33% of all trunk inclination stills were con-sidered impossible to rate by at least one of the three observers, and 46% of all stills of the upper arm [37]. Loss of data imply that the effective cost of exposure sampling will increase because longer periods of work must be assessed to reach the target efficiency. Since observa-tions–at least in the present occupation–appeared to be lost at a higher proportion than

inclinometer data, this may further increase the relative disadvantage of observations in the cost-efficiency comparison.

**Other measurement strategies.** In the present study, we did not simulate strategies with more than one repeated analysis per observer. Mathiassen et al. [33] compared the cost-efficiency of different observation strategies for measuring exposures among hairdressers. The strategies involved from 1 to 4 observers, each observing the same video stills 1 to 4 times, and the study found that the best strategy was to have 4 observers rate each still once. This may not be the optimal choice if the variance between observers is small and that within an observer is large, while the cost of engaging an observer is substantially larger than that of observing stills repeatedly [30], but we based the simulated observation strategies in the present study on that finding.

**Errors in cost and efficiency.** We used explicit estimates of costs and exposure variance components and did not include any considerations to uncertainties associated with these estimates. However, both are, to some extent, uncertain. The variance components were determined on basis of a limited sample of paper mill workers working for a limited number of days, and will, for that reason, be uncertain vis-à-vis the 'true' variance between and within workers. Notably, variance components require substantially more data to reach a stable estimate than, for instance, mean exposure values [64], and the between-observer variance in the present study, based on the output from three observers, is likely associated with a considerable uncertainty. Cost components were suggested in a previous investigation of paper mill workers to also be uncertain: the best- and worst-case cost scenarios for assessments of both trunk and arm inclination differed by a factor close to 6 [36]. Very few studies have addressed uncertainties in estimates of variance and cost, and no study has so far addressed the effects of these uncertainties on cost-efficiency. However, we acknowledge, as commented in the study by Waleh-Åström et al. [36], that even in the present setting, some realistic cost scenarios might lead to other conclusions than the almost uniform superiority of inclinometry suggested in the present cost-efficiency analysis.

## Future research on cost-efficiency

The present study is one of the first to present a comprehensive cost model that can be used in exposure data collection and processing in combination with the precision of the resulting mean exposure to inclination of the trunk and upper arm. Since this field of research is still in its infancy, further research is needed in several areas.

**More data on cost and efficiency.** First, data on both costs and exposure variability are still limited in the literature, and further studies should document cost components and sources of variance in new settings, e.g. office work, for new variables, e.g. neck postures, and for other data collection scenarios, e.g. that observations are performed only on-site [33] or are distributed between a number of observers rather than having every observer rating each video still. Descriptive data should then include estimates of uncertainty, both for costs and exposure variability.

**Cost-efficient estimation of costs.** Second, our comprehensive cost model (Eq 1) likely includes cost components that are small but costly to monitor. Neglecting those components would lead to cost estimates deviating weakly from those obtained with the complete cost model, while a more parsimonious assessment of costs would be considerably cheaper. Thus, more research is needed to identify options for obtaining a better 'cost-efficiency' of the cost assessment *per se*. Also, more diversified analyses of costs are needed, including, for instance, a discussion of 'opportunity costs', i.e. the lost opportunities for benefits when using a measurement strategy that is more costly, i.e. less cost-efficient, than what it could be.

**Non-standard data sets.** Finally, the present study as well as other studies of cost-efficiency are based on a number of assumptions regarding the study design, e.g. that data are balanced, and that no data are lost. Likely, these assumptions are violated to different extents in assessments of exposure in the field, and we propose research in different occupational settings that can increase the insight into cost-efficiency consequences of violated assumptions.

## Conclusions

In the present simulation illustrated by empirical data on paper mill workers, we found that inclinometry was, in general, more cost-efficient than observations when determining trunk and upper arm inclination. We compared measurement strategies differing in the number of subjects and shifts per subjects, and for observation the number of observers, and found that for both inclinometry and observation, arm inclination could be determined with better cost-efficiency than trunk inclination. Since we used a comprehensive cost model that could not be optimized, we reported cost-efficiency in empirical diagrams (cf. Fig 1), and we recommend future studies to use similar ways of communicating and comparing results of different data collection strategies. Our numerical data for costs and exposure variability are specific to paper mill work and the measurement strategies used by us, and we recommend future data collections to consider, in a pilot study, different options for data collection and processing, based on the conditions at hand. However, we believe that inclinometry will, in a far majority of occupational data collections, be more cost-efficient than observation for determining postures. This advantage of inclinometry will likely be even more pronounced in the future since both inclinometers and data collection protocols in inclinometry develop faster than observation techniques.

## Supporting information

**S1 File. Costs, source data.** The document gives a link to a previously published paper in Applied Ergonomics by the current authors, from which the source data for estimation of cost components (Table 1) can be downloaded.
(DOCX)

**S1 Dataset. Inclinometry, source data.** The table shows data at the level of individual workers, both for upper arm and trunk inclination. Within workers (n = 28), data are listed for each measured shift (n = 3). The table shows the data used for estimation of group means and variance components (Table 2) after eliminating shifts with more than 50% computer work (marked in red). Yellow cells mark shifts that were lost due to technical problems. All 28 workers are included in the table, even workers #1, #7, #14 and #15, who were completely disqualified.
(XLSX)

**S2 Dataset. Observation, source data.** The table shows data at the level of individual workers, both for upper arm and trunk observation. Within workers (n = 28), data are listed for each measured shift (n = 3), and within shifts for each observer (n = 3). The table shows the data used for estimation of group means and variance components (Table 2) after eliminating shifts with more than 50% computer work (marked in red). All 28 workers are included in the table, even workers #1, #7, #14 and #15, who were completely disqualified.
(XLSX)

## Acknowledgments

We would like to thank all involved staff at the investigated paper mill. We also appreciate the efforts made by Camilla Zetterberg, Mahmoud Rezagholi, Karin Holmkvist, Niklas Lindfors,

Magdalena Lindquist, Viktor Lyskov, Pontus Wiitavaara, Lena Liljedahl, Mikael Forsman, and Per Gandal in collecting and processing the data.

## Author Contributions

**Conceptualization:** Svend Erik Mathiassen, Amanda Waleh Åström, Annika Strömberg, Marina Heiden.

**Data curation:** Marina Heiden.

**Formal analysis:** Svend Erik Mathiassen, Amanda Waleh Åström.

**Funding acquisition:** Svend Erik Mathiassen, Marina Heiden.

**Investigation:** Svend Erik Mathiassen, Marina Heiden.

**Methodology:** Svend Erik Mathiassen.

**Project administration:** Svend Erik Mathiassen, Marina Heiden.

**Resources:** Svend Erik Mathiassen, Marina Heiden.

**Supervision:** Svend Erik Mathiassen, Annika Strömberg, Marina Heiden.

**Writing – original draft:** Svend Erik Mathiassen.

**Writing – review & editing:** Svend Erik Mathiassen, Amanda Waleh Åström, Annika Strömberg, Marina Heiden.

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
