## [Decision Letter · Decision Letter 0]

24 Aug 2023

PONE-D-23-09700Cost and statistical efficiency of posture assessment by inclinometry and observation, exemplified by paper mill workPLOS ONE

Dear Dr. Mathiassen,

Thank you for submitting your manuscript to PLOS ONE. After careful consideration, we feel that it has merit but does not fully meet PLOS ONE’s publication criteria as it currently stands. Therefore, we invite you to submit a revised version of the manuscript that addresses the points raised during the review process.

Generally acclimated as a practically ready for publication manuscript by our esteemed panel of reviewers, we do suggest some finishing touches and look forward to perusing your final manuscript. Please notice the reviewers feedback and suggestions to enhance the impact of your upcoming PLOS One article.

We look forward to receiving your revised manuscript.

Kind regards,

Denis Alves Coelho, PhD

Academic Editor

PLOS ONE

Journal Requirements:

2. We noted in your submission details that a portion of your manuscript may have been presented or published elsewhere. [Source data exemplifying the approach described for assessment of cost-efficiency in the present manuscript were taken from (1) two publications by Heiden et al. (references #17 and #23 in the present manuscript), (2) a publication by Waleh-Åström et al. (reference #36). The first author of the present manuscript co-authored all three publications. The present use of data does not constitute dual publication, since the two publications by Heiden et al. were merely descriptive regarding paper mill work, and the publication by Waleh-Åström et al. considered uncertainties in cost assessments. None of the three previous publications addressed cost-efficiency, and data are only used here as an illustration of our suggested approach for how to assess that. Thus, the information communicated by the present manuscript is entirely new.] Please clarify whether this [conference proceeding or publication] was peer-reviewed and formally published. If this work was previously peer-reviewed and published, in the cover letter please provide the reason that this work does not constitute dual publication and should be included in the current manuscript.

Additional Editor Comments:

The review process was lengthy but the outcome is very positive, thank you for your patience.

Reviewers' comments:

Reviewer's Responses to Questions

**Comments to the Author**

1. Is the manuscript technically sound, and do the data support the conclusions?

Reviewer #1: Yes

Reviewer #2: Yes

2. Has the statistical analysis been performed appropriately and rigorously? 

Reviewer #1: Yes

Reviewer #2: N/A

3. Have the authors made all data underlying the findings in their manuscript fully available?

Reviewer #1: No

Reviewer #2: No

4. Is the manuscript presented in an intelligible fashion and written in standard English?

Reviewer #1: Yes

Reviewer #2: Yes

5. Review Comments to the Author

Reviewer #1: Dear Authors,

I have finished my review on your paper. Indeed, it is quite interesting and well written. I have no major comments and suggestions. Perhaps the Authors would like to discuss in 1-2 sentences how the costs would be affected by running such data collection-processing tasks in other countries and how would new technologies such as AI affect their estimates. Indeed, the use of AI and computer vision to make accurate estimates on postures could be at the beginning but there are many examples of object and instance-based segmentation that could apply to postural checkups.

Best regards and good luck,

S.

Reviewer #2: This article is well-written, describing the relative advantages in terms of cost and efficiency of inclinometry and observation (using video technology) considering several levels of variation (number of worker, number of shifts, number of observers). The general premise is interesting and useful. References are excellent and extensive.

This article is worthy of publication with minor revisions. It will be a useful addition to the litterature.

It is unclear to me the limitations applying to data availability as stated by the authors. I assume this is because data may be considered proprietary having been the subject of previous publications, however ease of access to this data may yet be a concern for PLOS One rules.

I found the annotation sometimes difficult to follow, despite clear explanations in the written article, since the number of variables was large. Consider using more bullet points (lists) rather than paragraph-based descriptions.

It would be useful to include a second figure that consists of enlarged areas described on pages 12 and 13, with each datapoint explicitly labelled to facilitate direct visual interpretation (not simply based on the text descriptions which are not visible because of scale).

While I was easily convinced of the validity of equations 2 and 3 (a and b), equations 4 (Q, a and b) remained difficult to follow, causing me to doubt the subsequent interpretations. Consider further explanation in this article around those equations (rather than referencing other works exclusively).

Finally three minor points:

- On p. 13, "assuming normally distributed data" is stated but whether this is a valid assumption is not addressed. Please clarify.

- In the Discussion section, observation is assumed to be done manually, whereas recent developments allow angular estimation from a single camera view on a cellphone. Please include this element in your discussions, as you have done for inclinometry technological cost changes.

- On p. 19, use "custom built" rather than "customary built" for clarity.

6. PLOS authors have the option to publish the peer review history of their article (what does this mean?). If published, this will include your full peer review and any attached files.

Reviewer #1: No

Reviewer #2: No

---

## [Author Response · Author response to Decision Letter 0]

13 Sep 2023

Journal Requirements:

We have done our best to follow these templates

2. We noted in your submission details that a portion of your manuscript may have been presented or published elsewhere. [Source data exemplifying the approach described for assessment of cost-efficiency in the present manuscript were taken from (1) two publications by Heiden et al. (references #17 and #23 in the present manuscript), (2) a publication by Waleh-Åström et al. (reference #36). The first author of the present manuscript co-authored all three publications. The present use of data does not constitute dual publication, since the two publications by Heiden et al. were merely descriptive regarding paper mill work, and the publication by Waleh-Åström et al. considered uncertainties in cost assessments. None of the three previous publications addressed cost-efficiency, and data are only used here as an illustration of our suggested approach for how to assess that. Thus, the information communicated by the present manuscript is entirely new.] Please clarify whether this [conference proceeding or publication] was peer-reviewed and formally published. If this work was previously peer-reviewed and published, in the cover letter please provide the reason that this work does not constitute dual publication and should be included in the current manuscript.

We have included the requested information in the enclosed cover letter

We have described data availability in the enclosed cover letter. Accordingly, we have uploaded supporting information (three files) with a link to the source data for estimation of cost components published in Applied Ergonomics (Waleh-Åström et al. 2018) and two new tables showing data at the individual level for inclinometry and observation, respectively.

We have checked the reference list

Additional Editor Comments:

The review process was lengthy but the outcome is very positive, thank you for your patience.

It was lengthy, indeed… But thank you for having taken it this far 😊

Comments by Reviewer #1

Dear Authors,

I have finished my review on your paper. Indeed, it is quite interesting and well written. I have no major comments and suggestions. Perhaps the Authors would like to discuss in 1-2 sentences how the costs would be affected by running such data collection-processing tasks in other countries and how would new technologies such as AI affect their estimates. Indeed, the use of AI and computer vision to make accurate estimates on postures could be at the beginning but there are many examples of object and instance-based segmentation that could apply to postural checkups.

Best regards and good luck,

S.

We have added that costs are specific to Sweden, but we do not find it justified to discuss any general relationships between costs and efficiency in different countries:

p.18: While the explicit results of the present study (Fig 1) are specific to the investigated population and measurement strategies, and to the Swedish cost structure, they illustrate some general results that can be expected to apply even to other populations.

We have also included considerations regarding the future of data collection and costs, in addition to those that were already discussed in the first manuscript. We have specifically mentioned the likely (yet unknown) role of AI and computer vision technology:

p.20: However, technological developments may, in a more distant future, lead to automated procedures based on Artificial Intelligence (AI), e.g. for determining postures from single camera views in smart phones. This will change the cost-efficiency relationship between inclinometry and observation, and now to the advantage of the latter.

Comments by Reviewer #2

Reviewer #2: This article is well-written, describing the relative advantages in terms of cost and efficiency of inclinometry and observation (using video technology) considering several levels of variation (number of worker, number of shifts, number of observers). The general premise is interesting and useful. References are excellent and extensive.

This article is worthy of publication with minor revisions. It will be a useful addition to the litterature.

Thank you for your positive review!

It is unclear to me the limitations applying to data availability as stated by the authors. I assume this is because data may be considered proprietary having been the subject of previous publications, however ease of access to this data may yet be a concern for PLOS One rules.

We have uploaded supporting information (three files) with a link to the source data for estimation of cost components published in Applied Ergonomics (Waleh-Åström et al. 2018) and two new tables showing data at the individual level for inclinometry and observation, respectively.

I found the annotation sometimes difficult to follow, despite clear explanations in the written article, since the number of variables was large. Consider using more bullet points (lists) rather than paragraph-based descriptions.

We appreciate that the reviewer considered our explanations to be clear. In the previous manuscript, we already noted the definition of most variables in bullet point lists, but we have now revised the list of measurement units (page 7-8) to also appear in bullet points.

It would be useful to include a second figure that consists of enlarged areas described on pages 12 and 13, with each datapoint explicitly labelled to facilitate direct visual interpretation (not simply based on the text descriptions which are not visible because of scale).

This is an excellent idea, and we have now included two additional figures (fig 2a and 2b), showing enlarged sections of fig 1, so as to clearly illustrate the relationships described on p. 12-13 

While I was easily convinced of the validity of equations 2 and 3 (a and b), equations 4 (Q, a and b) remained difficult to follow, causing me to doubt the subsequent interpretations. Consider further explanation in this article around those equations (rather than referencing other works exclusively).

These equations follow standard statistical principles of how to use the results of specified variance component analyses, like those presented in equations 3a and 3b, for estimating the overall variance of a mean exposure. We find that it would require too much information to explain these basic principles, and that it would also lie beyond the purpose of the present study. We have, however, added a sentence explaining that our equations 4a and 4b for estimating statistical performance were developed according to standard statistical principles:

p.10: Equations 4a and 4b both follow from standard procedures for transforming the variance components in a specific study design (derived using equations 3a and 3b) into the expected statistical performance of a corresponding estimate of a group mean exposure [39].

… where reference #39 is a statistical textbook (Searle et al. (2006): ‘Variance Components’)

Finally three minor points:

- On p. 13, "assuming normally distributed data" is stated but whether this is a valid assumption is not addressed. Please clarify.

The cited passage in the manuscript refers to an illustrative example of the relative uncertainties of different measurement strategies at the same budget (€ 10000). Thus, normality is not explicitly assumed for the present data, but used as an example so that the reader can get an impression of how different data collection methods compare to one another (at the same cost). We have made that clear by revising the text:

p.13: If data were, as an example, normally distributed, these SDs…

- In the Discussion section, observation is assumed to be done manually, whereas recent developments allow angular estimation from a single camera view on a cellphone. Please include this element in your discussions, as you have done for inclinometry technological cost changes.

As commented in the answer to reviewer #1, we have added a sentence in the discussion, in the context of comparing cost-efficiency between inclinometry and observation:

p.20: However, technological developments may, in a more distant future, lead to automated procedures based on Artificial Intelligence (AI), e.g. for determining postures from single camera views in smart phones. This will change the cost-efficiency relationship between inclinometry and observation, and now to the advantage of the latter.

- On p. 19, use "custom built" rather than "customary built" for clarity.

We have revised the text as proposed.

---

## [Editor Report · Decision Letter 1]

18 Sep 2023

Cost and statistical efficiency of posture assessment by inclinometry and observation, exemplified by paper mill work

PONE-D-23-09700R1

Dear Dr. Mathiassen,

We’re pleased to inform you that your manuscript has been judged scientifically suitable for publication and will be formally accepted for publication once it meets all outstanding technical requirements.

Kind regards,

Denis Alves Coelho, PhD

Academic Editor

PLOS ONE

---

## [Editor Report · Acceptance letter]

25 Sep 2023

PONE-D-23-09700R1 

Cost and statistical efficiency of posture assessment by inclinometry and observation, exemplified by paper mill work 

Dear Dr. Mathiassen:

I'm pleased to inform you that your manuscript has been deemed suitable for publication in PLOS ONE. Congratulations! Your manuscript is now with our production department. 

Kind regards, 

on behalf of

Dr. Denis Alves Coelho 

Academic Editor

PLOS ONE